# Lesion Segmentation Framework Based on Convolutional Neural Networks with Dual Attention Mechanism

**Fei Xie [1,2,†], Panpan Zhang [3,†], Tao Jiang [4], Jiao She [3], Xuemin Shen [5,\*], Pengfei Xu [3,\*], Wei Zhao [6], Gang Gao [7] and Ziyu Guan [6]**

1 School of AOAIR, Xidian University, Xi'an 710075, China; fxie@xidian.edu.cn
2 Department of Electronic Engineering, Tsinghua University, Beijing 100084, China
3 School of Information Science and Technology, Northwest University, Xi'an 710069, China; 201932137@stumail.nwu.edu.cn (P.Z.); shejiao@stumail.nwu.edu.cn (J.S.)
4 Zhejiang Provincial Seaport, Ningbo 315040, China; jiangt@nbport.com.cn
5 Department of Oral Mucosal Diseases, Shanghai Ninth People's Hospital, College of Stomatology, Shanghai Jiao Tong University School of Medicine, Shanghai 20025, China
6 School of Computer Science and Technology, Xidian University, Xi'an 710075, China; ywzhao@mail.xidian.edu.cn (W.Z.); zyguan@xidian.edu.cn (Z.G.)
7 Shaanxi Great Wisdom Medical Care Technologies Co., Ltd., Xi'an 710075, China; xiefei@nwpu.edu.cn
\* Correspondence: SHENXM1327@sh9hospital.org.cn (X.S.); pfxu@nwu.edu.cn (P.X.)
† They are the Co-first author.

**Abstract:** Computational intelligence has been widely used in medical information processing. The deep learning methods, especially, have many successful applications in medical image analysis. In this paper, we proposed an end-to-end medical lesion segmentation framework based on convolutional neural networks with a dual attention mechanism, which integrates both fully and weakly supervised segmentation. The weakly supervised segmentation module achieves accurate lesion segmentation by using bounding-box labels of lesion areas, which solves the problem of the high cost of pixel-level labels with lesions in the medical images. In addition, a dual attention mechanism is introduced to enhance the network's ability for visual feature learning. The dual attention mechanism (channel and spatial attention) can help the network pay attention to feature extraction from important regions. Compared with the current mainstream method of weakly supervised segmentation using pseudo labels, it can greatly reduce the gaps between ground-truth labels and pseudo labels. The final experimental results show that our proposed framework achieved more competitive performances on oral lesion dataset, and our framework further extended to dermatological lesion segmentation.

**Keywords:** medical image segmentation; computational intelligence; convolutional neural networks; weakly supervised segmentation; attention mechanism





## 1. Introduction

With the rapid development of computer vision, especially the significant improvement of the representation ability of convolutional neural networks [1,2], image segmentation has achieved good performances and laid a solid foundation for the application of medical image segmentation. Medical images segmentation as an important and difficult task of computer-aided diagnosis, is the key to further obtain diagnostic information. Traditional object location in medical images requires professional doctors to manually identify, which is not only time-consuming and labor-intensive but also vulnerable to subjective factors. While the lesion segmentation results obtained by deep learning methods are now becoming a promising method. However, compared with ordinary images, clinical diagnosis invokes higher requirements for the accuracy of the segmentation results of medical images. In addition, the high variability, the complex morphological structure,

the ambiguity and the scarce labels of lesions in medical images pose great challenges to medical image segmentation [3].

Recently, lesion segmentation methods based on deep convolutional neural networks have been widely applied to medical image segmentation. Encoder-Decoder, FCNs [4] (Fully Convolutional Networks for Semantic Segmentation) and the methods based on extended convolutional neural network have become the mainstream segmentation methods. For example, U-Net designed in [5] an "U-shaped" network, and symmetric expansion paths are added to enhance the positioning representation capability of the network. U-Net is superior to the previous methods in terms of the amount of data required, the efficiency and accuracy of methods. Since then, more and more variants of U-Net [6–11] are proposed to enhance the network presentation capabilities, the transmission and fusion of feature information within and between layers to further improve the segmentation accuracy. U-net and its variants perform well in medical images such as CT(Computed Tomography) and MRI(Magnetic Resonance Imaging). On the one hand, the CT and MRI images are mostly single-channel grayscale images, with simple semantics and relatively fixed structures. On the other hand, the U-Net network has fewer parameters, and the skip connection of U-net plays an important role. The skip connection can make the feature graph of the corresponding position of the encoder fuse on the channel in the up-sampling process of each level of the network. Through the fusion of low-level features and high-level features, the network can retain more high-resolution details contained in high-level feature images, thus improving the accuracy of image segmentation, so as it is not easy to overfit for relatively small medical datasets. Therefore, when there are relatively small medical datasets, this U-NET model is preferred to avoid overfitting.

There are also some medical image datasets consisting of visible images, such as our oral leukoplakia dataset and the ISIC2018 [12–14] used in this paper. Different from radiographic images such as CT and MRI images, this type of medical image taken by conventional visible light cameras have larger size, and relatively richer semantic information, while they also have challenges in terms of object segmentation. As shown in Figure 1, the same category of objects has some differences in visual features, while the features of different categories of objects have similarities. The texture, color, shape, and size of lesions in the images varies, and the boundary of lesions is blurred. In addition, all of the artifacts during image capturing, light intensity and reflections, bubbles, hair occlusion, background boards, and so forth, bring many difficulties to the segmentation task. Specifically, for the oral leukoplakia dataset built in this paper, the difficulties of leukoplakia lesion segmentation mainly lie in the morphology diversity of lesion, including granular, crumpled, warty, and so forth. In addition, the differences in the size of lesions, the blur boundaries between the lesions and their surrounding tissues, and the changeable locations of lesions, and so forth, will also increase the difficulty in the segmentation of leukoplakia. At present, there are few related works to oral lesion segmentation. Camalan et al. [15] developed a image classification method to identify the "suspicious" oral dysplasia or "normal" oral images through transfer learning on Inception-ResNet-V2. Jubair et al. [16] proposed a method to predict oral cancer from oral images using a lightweight transfer learning model. These methods are designed for the image classification task, while our method is performed for segmenting the oral lesion, which is the image segmentation task with more complexity. Figure 1 shows some examples of segmentation results of oral leukoplakia and skin disease lesions. All of these complex medical images pose big challenges for lesion segmentation, since the category filling rate loss in traditional image segmentation models often loses its effects of medical image segmentation.

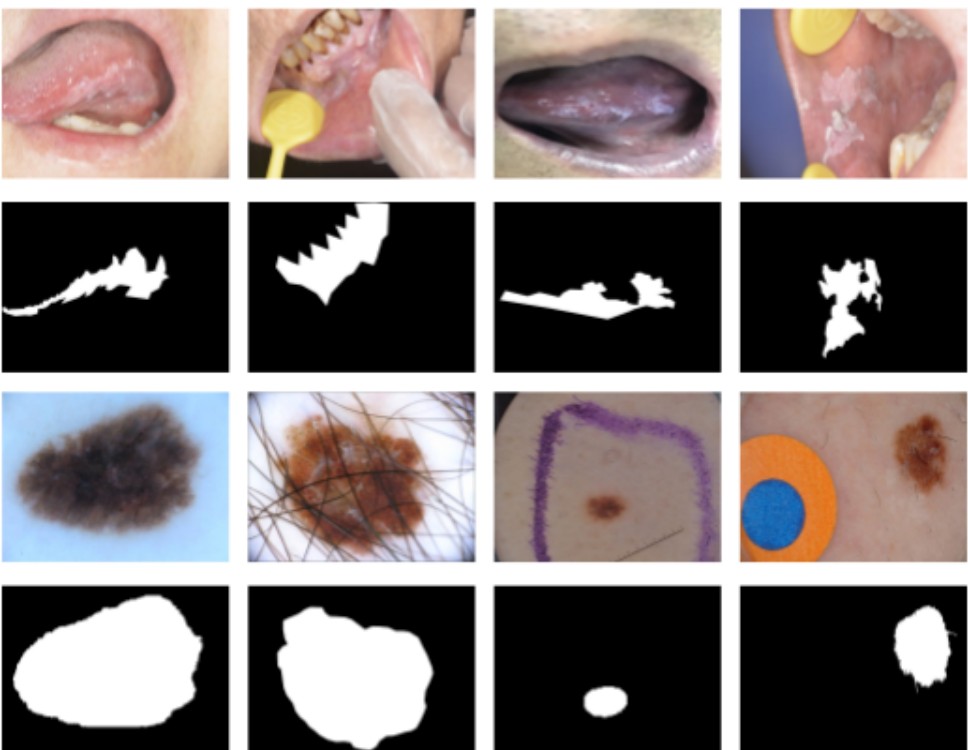

**Figure 1.** Some samples of lesion segmentation (The first two rows are oral leukoplakia dataset, and the last two rows are ISIC 2018 dataset).

For some types of medical images with extremely complex morphological and multi-scale features, traditional U-Net can not deal with the multi-scale features of the objects well, which makes it difficult for U-Net to extract the effective visual features of small objects. This results from the fact that the models have significant performance degradation. In order to solve these problems, it is particularly important to improve the network's visual feature capturing capabilities for medical images. Fortunately, many attempts have been made, and introducing the attention mechanism into different deep networks is a feasible direction. Compared with the U-Net, the structure of Mask R-CNN is more complex, especially the FPN backbone network. This network can adapt to multi-scale changes of lesions and extract effective regional features. Therefore, in this paper, we introduce a dual attention mechanism into Mask R-CNN [17–20], and propose a network to extract the effective visual features of lesions. The dual attention mechanism can help the network pay attention to feature extraction from important regions, which can improve the representation ability of the convolutional networks for lesion areas. The experimental results show that the models with the dual attention mechanism have the optimal segmentation boundaries, and fewer missed or false segmentation areas. Although it is possible to obtain better segmentation results through fully supervised learning, it requires fine pixel-level labels of the objects in images. Therefore, a professional pathologist is required to give labels for medical images, which can consume some economic and time and greatly limits the practical application of intelligent assistance systems. To solve this problem, the segmentation methods based on weakly supervised learning can use image-level and box-level coarse-grained labels to train a pixel-level fine segmentation network. In this paper, we make full use of our network with a dual attention mechanism, and integrate a weakly supervised segmentation branch. This improvement achieves the weakly supervised lesion segmentation, which has much less cost in image labeling. Furthermore, this segmentation framework can also be applied to radiomics lesion segmentation in the future.

Main contributions can be concluded as follows:

1.  In this paper, the researchers construct an end-to-end medical lesion segmentation framework, which has both fully supervised segmentation and weakly supervised segmentation branches. If pixel-level labels are used for the images, the fully supervised segmentation branch can be used for lesion segmentation. In the process of experiments if we only have box-level labels similar to the labels for object detection, the researchers can use the weakly supervised segmentation branch to achieve accurate lesion segmentation with comparable results to those obtained by the fully supervised segmentation methods.

2.  To solve the problem of inaccurate segmentation of lesion boundaries in the lesion segmentation task, the researchers introduce the CBAM [21] attention mechanism into the Mask R-CNN to help the network pay attention to fine-gain feature learning from the regions of interest. This improvement will be beneficial for the segmentation results, especially for the segmented lesion boundaries.

This paper is organized as follows. In Section 2, we first state some classical methods of fully and weakly supervised image segmentation (Sections 2.1 and 2.2), as well as the basic principles of attention mechanisms (Section 2.3). Then, the proposed image segmentation framework is described in Section 3, and we specifically give a statement for weakly supervision segmentation in Section 3.2. The validation experiments and analysis of the model are described in Section 4; here, we carried out several related methods and our improved method on public skin image datasets and our own oral leukoplakia image dataset. While the final conclusion is given in Section 5.

## 2. Materials and Methods

### 2.1. Fully Supervised Segmentation Methods

Fully supervised segmentation is divided into semantic segmentation and instance segmentation. In terms of semantic segmentation, since FCN [22] first introduced full convolutional neural networks into segmentation, a series of improved or redesigned segmentation methods [23–29] following this paradigm have achieved good results. Currently, models based on the encoder-decoder structure have gradually become the mainstream segmentation framework such as SegNet [30,31], U-Net [5] and RefineNet [32]. The main reason is that this model can extract long-distance semantic information. In addition, ParseNet [33], DeepLabv2 [34], PSPNet [35] and other models based on spatial pyramid pooling [36] to capture long-distance contextual semantic features are becoming popular as well. In addition, among the methods mentioned above, Refine Net is a good network model. This model is a multipath optimized network for high resolution semantic segmentation. It makes perfect use of all available information in the downsampling process to achieve high resolution prediction of long-distance residual connections. Moreover, a network structure for generating high rate segmentation graph is provided by combining high level semantic features with low level features. This feature makes it suitable for multi-class semantic segmentation tasks. Compared with semantic segmentation, instance segmentation also needs to distinguish different instances of the same class of targets on the basis of semantic segmentation. Many methods [37–42] incorporating the region proposal network [43] (RPN) have achieved satisfying results. These methods first obtain the detection box of the target and then use another segmentation branches to segment the instance.

Among the new methods, Mask-RCNN [18] adds a mask branch based on faster R-CNN [44]. This branch changes ROI pooling to ROI alignment, so as to obtain pixel-level mask prediction. It also has the functions of object detection and instance segmentation. This method has become a general framework.

### 2.2. Weakly Supervised Segmentation Methods

The fully-supervised segmentation model can segment accurate results after training with a large number of pixel-wise labels, but it is extremely expensive to obtain pixel-wise la-

bels. To address this issue, different levels of weakly supervised labels are adopted to solve the problem of manually labeling large amounts of data, such as image-level [39,45–50], scribbles [51–53], and point labels [54]. Because these weakly supervised labels provide limited prior information, it is difficult to produce satisfactory results for the complex medical images. In this article, we focus on using bounding box-level labels to balance labeling cost and segmentation accuracy. Previous box-level weakly supervised segmentation methods usually need to manually generate weakly supervised pseudo labels, and then use pseudo labels for training on fully supervised methods. Specifically, it can be divided into three stages: the first stage uses GrabCut [55] or MCG [56] to generate pixel-level pseudo labels, the second stage uses the generated pseudo labels as ground-truth to train the segmentation model, and the third stage uses an iterative algorithm or conditional random fields (CRFs) [2,57–60] to optimize and post-process the segmentation results. Therefore, it is difficult to solve the gap between the ground-truth and the pseudo label, and the final segmentation result will be significantly worse than the effect of the pseudo label. Compared with previous work, our weakly supervised segmentation method is different in the following respects. Firstly, our model does not require manual generation of pseudo labels. Secondly, we first use box labels to train the target detection model, and then use GrabCut [55] to separate the foreground and background regions of the detected target region in the inference stage. Thirdly, in terms of final mask optimization, we abandoned the use of CRFs [2,57–60], and instead adopted the faster ConvCRF [20,61–63] for post-processing.

### 2.3. Attention Mechanism

Human vision can quickly scan the global image to obtain the target areas that need to be focused on. The attention mechanism in deep learning is similar to the human visual attention mechanism, and the goal is to select the critical information in the current task. By adjusting the feature map, Wang proposed a residual attention network [64,65], which not only performs better but also is robust to noisy input. Oktay proposed the Attention-Unet [6,66] to suppress the information of unimportant regions, which is better in segmentation. Hu [67–69] proposed the Squeeze-and-Excitation module based on the relationship between channels. It only uses global average pooling to calculate channel attention. However, as shown in spatial attention[70], it also plays an important role in convolutional networks. It will tell the network "where" to focus. Since then, the application of channel attention and spatial attention have become a consensus. DANet [71] introduces the idea of self-attention, which can better capture features through a long-range context. CCNet [72,73] skillfully uses the criss-cross idea, which greatly reduces the amount of calculation. In this paper, we add CBAM [21] to Mask R-CNN [18] for the first time, which also has channel attention and spatial attention. Channel attention tells us "what" is meaningful and spatial attention tells us "where" important information is.

### 3. Methods

The Mask R-CNN is extended from Faster R-CNN [18,74] and is a two-stage framework. The first stage is the Region Proposal Network [43] (RPN). In the second stage, further fine-tuning frames for the ROI proposed by RPN. Finally, the parallel Mask head branch will segment the target mask.

In the weakly supervised segmentation branch, we directly use the detection boxes to segment the lesion area, to avoid the gap between the pseudo labels and ground-truth labels. This improves the segmentation performance. The key points and differences of the method are detailed in the following subsections. The architecture of the segmentation model is shown in Figure 2.

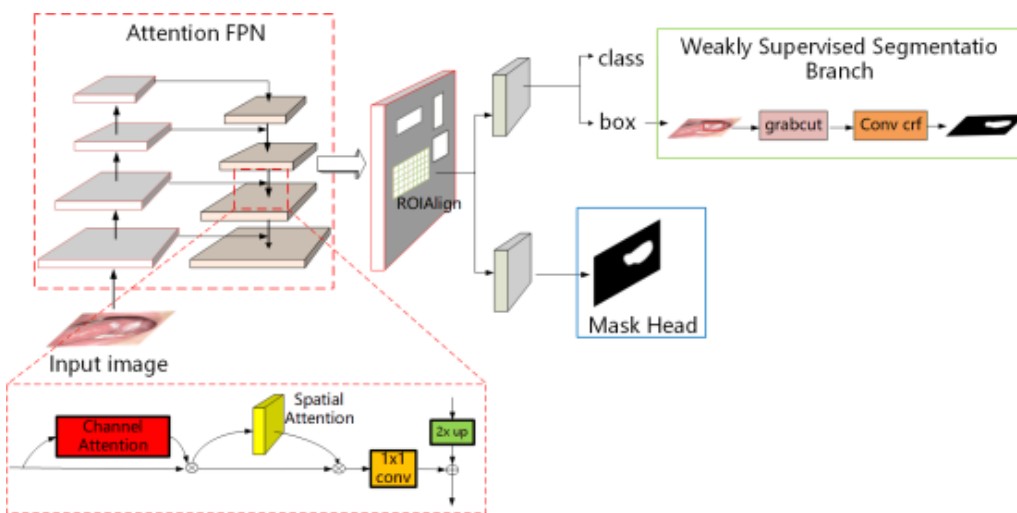

**Figure 2.** Medical lesion segmentation framework based on dual attention mechanism.

### 3.1. Segmentation Model Based on Dual Attention Guidance

FPN, as the backbone network of Mask R-CNN, performs well in conventional segmentation tasks. However, when segmenting medical images with complex and fuzzy boundaries, it often results in missing segmentation or wrong segmentation. The reason is that the correct features of the lesion area are not extracted. Therefore, we add the dual attention module in the backbone network and the improved attention-FPN structure is shown in the attention FPN part of Figure 2. As shown in Figure 3, the specific joining position of the dual attention module is in the Conv block and identity block of ResNet. Conv block and identity block are the basic modules of ResNet network. Conv block has convolution operation on branches, which can change the number of output channels of the block; Identity block has no operation on the branch, and the number of input and output channels of this block is the same. In our model, a dual attention mechanism is added to these two different blocks.

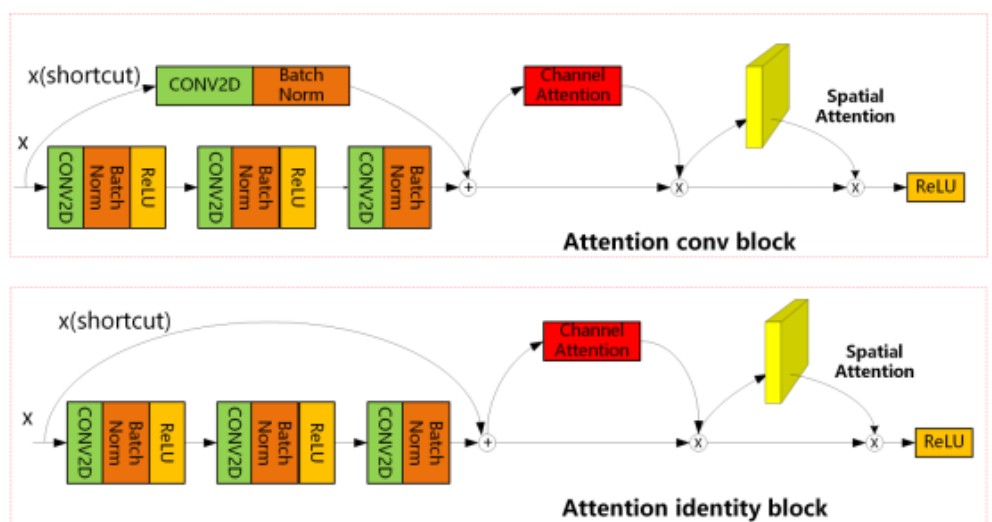

**Figure 3.** Attention conv block and Attention identity block.

In CNN, if the sizes of the convolution kernels are smaller than the step size, the performance of detection and segmentation will decline linearly. FPN is a clever solution by up-sampling high-level features and top-down connecting of low-level features. In the feature extraction and fusion stage, FPN performs well, especially for small target detection. We then use ResNet [75] series in the backbone network to have more flexible choices. Taking ResNets as the backbone network, the FPN network contains three paths, a bottom-

up path, a top-down path, and a horizontal connection in the middle. In the forward propagation process of CNN, the feature map after the calculation of the convolution kernel is usually small. The size of the feature map will change after passing through some layers, but not for the other layers. The layers that do not change the size of the feature map are classified as a stage. Specifically, for ResNets, the feature output of the last residual block of each stage is used to activate the output. For conv2, conv3, conv4, and conv5 outputs, the outputs of these final residual blocks are represented as C2, C3, C4, C5, and they have a step size of 4, 8, 16, 32 relatively to the original input image. The top-down process is to perform two times the up-sampling of higher-level features with more abstract and stronger semantic information, and to merge the output results of the up-sampling with the feature map of the previous layer generated from the bottom up through the horizontal connection. After fusion, the high-level features have been strengthened, and the two horizontally connected features should have the same spatial size. This is done to make use of the positioning details of the bottom layer. Each feature map is composed of many channels. In Mask R-CNN, the outputs of ResNet C2, C3, C4, C5 are passed to the next layer of the network for fusion. All channels of these output feature maps are given the same weight, that is, the same attention, but some of these channels are meaningless or erroneous features. We use channel attention after C2, C3, C4, C5 to capture the relationship among global channels. In other words, it encodes different weights for each channel to enhance the weight of important channels and suppress the features of unimportant channels.

In order to calculate the channel attention, the spatial dimensions of the input feature are compressed, the global maximum pooling and global average pooling are performed respectively, and then the multilayer perceptron model (MLP) output features are added and operated through the shared MLP. As shown in Figure 4a, after sigmoid activation operation, the final channel attention map is generated. Multiply the channel attention map and the input feature to generate the input feature of the spatial attention module.

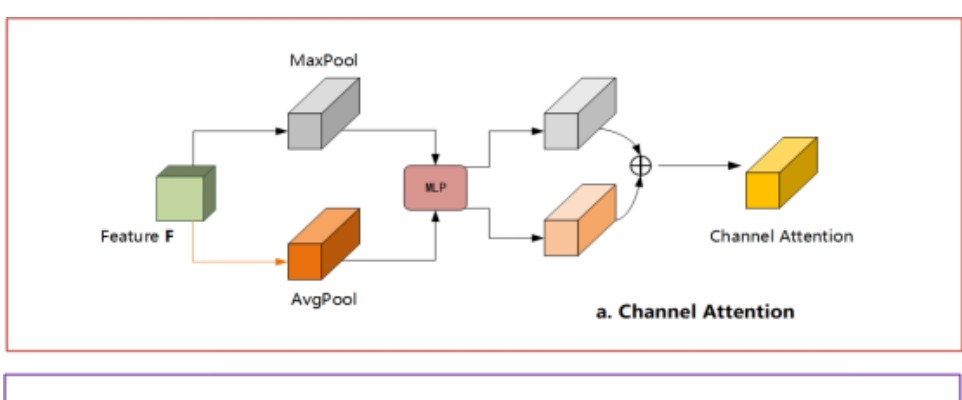

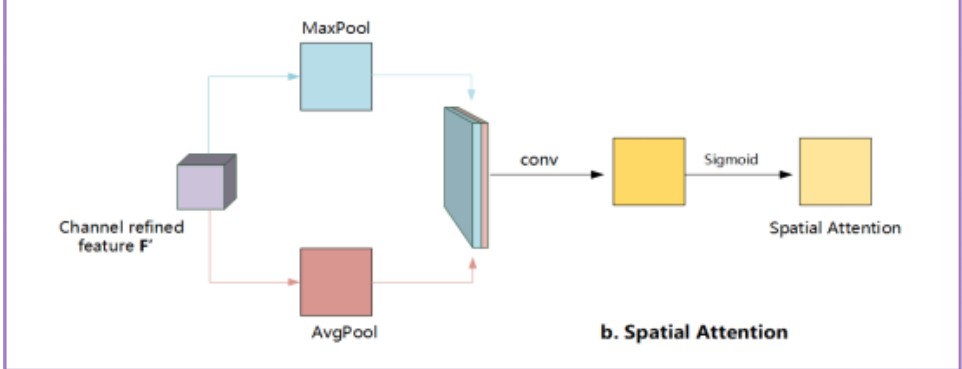

**Figure 4.** The overall structure of dual attention ((**a**). Channel Attention; (**b**). Spatial Attention).

The calculation process is as follows:

$$
\begin{aligned}
\mathbf{M_c}(\mathbf{F}) &= \sigma(\text{MLP}(\text{AvgPool}(\mathbf{F})) + \text{MLP}(\text{MaxPool}(\mathbf{F}))) \\
&= \sigma\left(\mathbf{W}_1\left(\mathbf{W}_0\left(\mathbf{F}_{\text{avg}}^{\text{c}}\right)\right) + \mathbf{W}_1(\mathbf{W}_0(\mathbf{F}_{\text{max}}^{\text{c}}))\right)
\end{aligned}
\tag{1}
$$

In the above formula, $\mathbf{F_{avg}^c}$ and $\mathbf{F_{max}^c}$ represent the averaged pooling feature and the maximum pooling feature, respectively, and $\sigma$ represents the sigmoid activation function. $M^c \in R^{c \times 1 \times 1}$, $W_0$ and $W_1$ are the weights of MLP.

In the original FPN network, the bottom-up and top-down features are fused directly into the horizontal connection, which lacks the spatial dependence among pixels. We use the spatial relationship among pixels to generate a spatial attention map, making the network pay attention to "where" the information is, which supplements channel attention.

In order to calculate the spatial attention, we use the feature map output by the channel attention module as input to perform global maximum pooling and global average pooling on the channel axis, respectively. We then conduct the concat operation and the $7 \times 7$ convolution operation. Finally, the $1 \times H \times W$ spatial attention map is generated through the sigmoid activation function, as shown in the Figure 4b. The final feature map is obtained by multiplying the spatial attention graph and the input features of this module. Spatial attention is calculated as follows:

$$
\begin{aligned}
\mathbf{M_s}(\mathbf{F}) &= \sigma\left(f^{7 \times 7}([\,\text{AvgPool}(\mathbf{F}); \text{MaxPool}(\mathbf{F})])\right) \\
&= \sigma\left(f^{7 \times 7}\left(\left[\mathbf{F}_{\text{avg}}^{\text{s}}; \mathbf{F}_{\text{max}}^{\text{s}}\right]\right)\right).
\end{aligned}
\tag{2}
$$

In the above formula, $\mathbf{F_{avg}^s}$ and $\mathbf{F_{max}^s}$ represent the averaged pooling feature and the maximum pooling feature respectively, and the dimension is $1 \times H \times W$. $\sigma$ represents the sigmoid activation function.

The outputs of ResNet C2, C3, C4, C5 calculate the one-dimensional channel attention $M_c \in R^{c \times 1 \times 1}$ on the channel axis through channel attention module and a two-dimensional spatial attention map $M_s \in R^{1 \times H \times W}$ is calculated on the spatial axis through the Spatial Attention module. Then, the final feature map is calculated through the series connection and the process is shown in Figure 3. The training of the fully supervised segmentation network is carried out using a dataset with pixel-level labels. The training process is the same as the original Mask R-CNN. The detection branch and the Mask Head branch will be trained at the same time; the inference phase will generate the final segmentation result in the Mask Head branch.

### 3.2. Weakly Supervised Segmentation

In this section, we will show the weakly supervised segmentation method. Given a dataset $D = \{I_n, B_n\}_n^N$ with bounding box labels, $N$ represents the number of samples datasets, $I_n$ represents the $n$ th picture, and $B_n$ represents the box-level label of $I_n$. In this section, our goal is to build an end-to-end weakly supervised segmentation model using only box-level dataset D. We know that Mask R-CNN [18] not only has the function of instance segmentation but also has the ability of target detection. In Section 3.1, the improved segmentation model also performs well on target detection, which is an important merit for our weakly supervised segmentation method. Apart from the fully supervised segmentation framework explained before, we abandon the fully supervised mask head branch and add a weakly supervised segmentation branch (Figure 2). The overall segmentation framework is shown in Figure 2. In the inference stage, the detection branch will give the target's tight bounding box, the area outside the bounding box is the background area, and there are some background pixels mixed in the target box. We use GrabCut [55,76,77] to separate the foreground and background in the boundary box. So far, we have obtained preliminary segmentation results, but the foreground segmented by GrabCut [55,78] has

holes in the interior, and inaccurate boundaries. In order to obtain better performance, we use the faster ConvCRF [20,61–63] to generate the final segmentation mask.

When using GrabCut to separate the foreground and background according to the detection bounding boxes of the lesion area, the efficiency will be slow with large input image. In order to increase the speed, the image must be zoomed, but the small lesion area will lose a lot of information after the image size reduction, thus, finding a balance between efficiency and effect is necessary. In order to calculate the scale of the zooming, we determine the relative scale of the detection box and the image.

The training process of the weakly supervised segmentation network is different from the fully supervised process. The latter uses the box-level weakly supervised label dataset and only trains and updates the parameters of the detection branch, which is essentially the process of training a target detection network. In the inference stage, the weakly supervised segmentation branch will generate weakly supervised segmentation results based on the output bounding box of the target detection network.

## 4. Result

### 4.1. Experimental Details and Evaluation Strategies

Experimental details: The proposed method is evaluated on two popular datasets including the OLK dataset and the ISIC [12] 2018. We use the keras framework to implement our model. We use ResNets as our backbone network and fine tune the network from a pre-trained model which is learned on the MS COCO dataset. The batch size, learning rate, weight decay, momentum and Epoch are 2, 0.001, $10^{-4}$, 0.9 and 60, respectively. The optimizer is Adam, and data enhancement, such as rotation, affine transformation, and random clipping, are performed. The framework is trained on a machine with a NVIDIA TITAN RTX 24 GB GPU.

Evaluation strategy: Like most medical image segmentation evaluation strategy, we use the standard F1-score (F1), sensitivity (SEN), specificity (SPE), accuracy (ACC), and Jaccard similarity to evaluate our proposed model.

### 4.2. Oral Leukoplakia Dataset

Oral leukoplakia is an injury to the oral mucosa and a precancerous lesion. We obtained the oral leukoplakia medical image dataset from the hospital which contains 90 original images and corresponding masks labeled by professional doctors. We divided the whole image dataset into a training set (55 images), a validation set (15 images), and a test set (20 images). Since the number of oral leukoplakia datasets is small, there is no test set. Compared with the ISIC 2018 dataset, the segmentation task of the oral leukoplakia dataset is more challenging. Not only is the number sparse—only 3% of the ISIC 2018 dataset—but also the boundary of the lesion area is more blurred, the shape is irregular and changeable. In the fully supervised segmentation experiment, the ground-truth labels are the binary masks of the original dataset. In the weakly supervised segmentation experiment, the ground-truth labels are the circumscribed rectangles of the binary masks.

Figure 5 shows the segmentation results of our proposed fully supervised and weakly supervised methods on the oral leukoplakia dataset. The results obtained by our fully supervised segmentation method basically remain consistent with the shapes of the ground truths, although the boundaries do not have very good consistency. Att-Deeplab-V3+ can achieve good performance in quantitative evaluation indicators, but it does not have good shape preservation of lesions with the ground truths, and so are the results obtained by Mask RCNN. In addition, our weakly supervised (WS) segmentation method can achieve segmentation results with more overlaps with the lesions in the ground truths. Table 1 shows the quantitative evaluation indicators of all the methods. From the experimental results, we can see that the proposed fully supervised (FS) and weakly supervised (WS) segmentation methods achieve the best performance, and have many improvements over the baseline method (Mask RCNN). In addition, the segmentation performance of the weakly supervised method is very close to that of the fully supervised method. Therefore,

the new weakly supervised segmentation model greatly reduces the cost of data annotation for the localization and segmentation of disease regions.

**Table 1.** Performance comparison of the proposed segmentation network and other methods on the Oral leukoplakia dataset.

| | Method | F1 | SEN | SPE | ACC | Jaccard Similarity |
|---|---|---|---|---|---|---|
| FS | Att-Deeplab v3+ [79] | 0.514 | 0.521 | 0.953 | 0.935 | 0.935 |
| | U2-Net [80] | 0.759 | 0.734 | 0.986 | 0.967 | 0.967 |
| | Mask R-CNN [18] | 0.741 | 0.704 | 0.978 | 0.959 | 0.959 |
| | Ours-full | **0.815** | **0.758** | **0.990** | **0.967** | **0.967** |
| WS | Ours-weak | **0.684** | **0.843** | **0.964** | **0.943** | **0.943** |

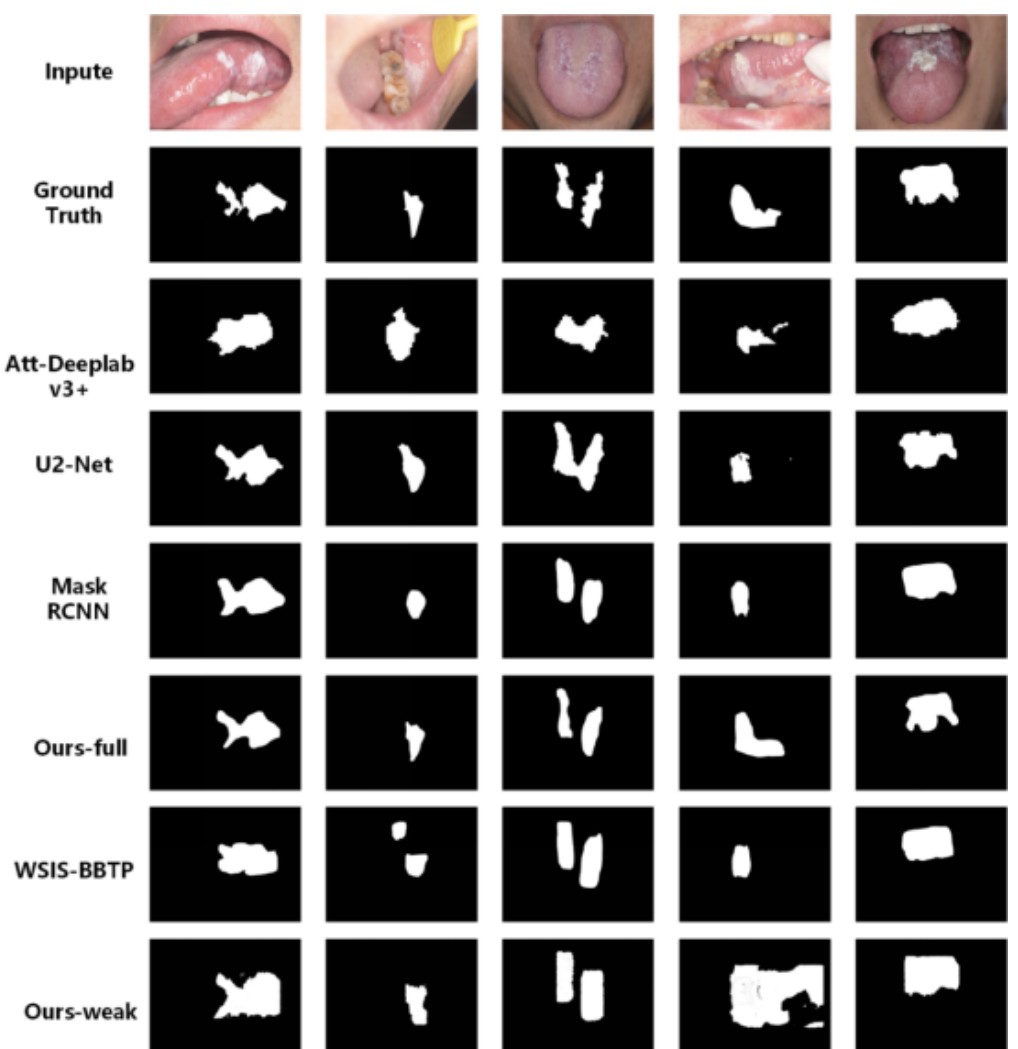

**Figure 5.** Segmentation results of fully supervised and weakly supervised segmentation method on the Oral leukoplakia dataset.

*4.3. ISIC*

The ISIC 2018 [12] challenge dataset was published by the international skin imaging collaboration (ISIC) in 2018. We select the dermatoscopy image lesion boundary segmentation dataset of challenge task 1, which contains 2594 original images and the corresponding binary ground-truth masks. In order to compare with other methods, we set up the same set with other methods, including 1815 training sets, 259 verification sets and 520 test sets. We

set the image input size to 768 × 768. In the fully supervised segmentation experiment, the ground-truth labels are the binary masks of the original dataset. In the weakly supervised segmentation experiment, the ground-truth labels are the circumscribed rectangles of the binary masks. In addition, ISIC 2017 is also a famous skin image dataset similar to ISIC 2018, we also conducted relevant experiments on ISIC 2017 to verify the performance of different models.

Figure 6 shows the segmentation results of the proposed fully supervised and weakly supervised methods on the ISIC 2018 [12] dataset. It is not difficult to find out that some of the methods suffer a serious performance degradation on the oral leukoplakia dataset, but achieved much better performance on the ISIC 2018 dataset. The main reason may be that the lesion regions in the skin disease images are easier segmented than those in oral leukoplakia images. Therefore, there is little difference in the result images by the image segmentation methods. In addition, the weakly supervised segmentation methods also achieved good performance, and their results are almost close to the results obtained by the fully supervised segmentation methods. However, the results of our weakly supervised segmentation method have better shape consistency with the ground truths than WSIS-BBTP. Furthermore, the quantitative evaluation indicators of the experimental results are shown in Table 2. It can be seen that the proposed fully supervised segmentation method achieved great improvements over the original mask R-CNN, and also achieves competitive results compared with other methods. At the same time, our weakly supervised segmentation method has better performance than WSIS-BBTP, and also achieved comparable performance regarding fully supervised segmentation methods, and even surpassed some fully supervised segmentation methods, such as U-net [5], Att U-net [6], R2U-net [81], Att R2U-Net [81], BCDU-Net [82]. Furthermore, the experimental results on ISIC 2017 are shown in Table 3. It can be seen that there have been better performances for different models compared with ISIC 2018, and our weakly supervised segmentation method also achieved a competitive performance with other fully supervised segmentation models.

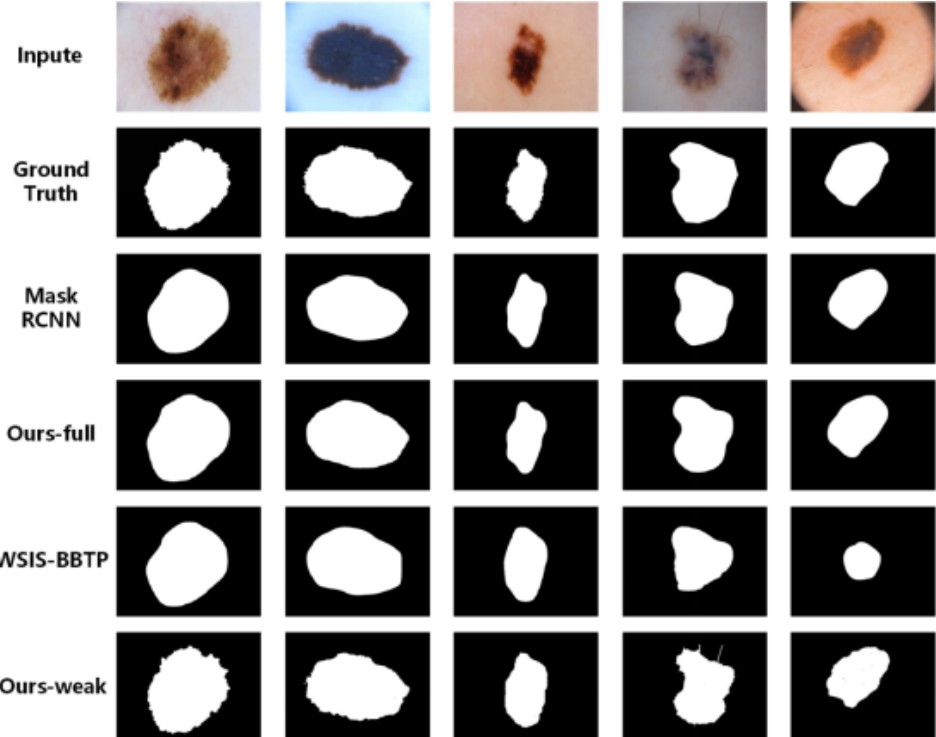

**Figure 6.** Segmentation results of fully supervised and weakly supervised segmentation methods on the ISIC 2018 dataset.

**Table 2.** Performance comparison of the proposed segmentation network and other methods on the ISIC 2018 dataset.

|    | Method | F1 | SEN | SPE | ACC | Jaccard Similarity |
|----|--------|-----|------|------|------|--------------------|
| FS | U-net [5] | 0.647 | 0.708 | 0.964 | 0.890 | 0.549 |
|    | Att U-net [6] | 0.665 | 0.717 | 0.967 | 0.897 | 0.566 |
|    | R2U-net [81] | 0.679 | 0.792 | 0.928 | 0.880 | 0.581 |
|    | Att R2U-Net [81] | 0.691 | 0.726 | 0.971 | 0.904 | 0.592 |
|    | BCDU-Net [82] | 0.851 | 0.785 | 0.982 | 0.937 | 0.937 |
|    | MCGU-Net [83] | 0.895 | 0.848 | 0.986 | 0.955 | 0.955 |
|    | Deeplab v3+ [25] | 0.882 | 0.856 | 0.977 | 0.951 | 0.951 |
|    | Att-Deeplab v3+ [79] | 0.712 | 0.875 | **0.988** | **0.964** | **0.964** |
|    | Mask R-CNN [18] | 0.872 | 0.846 | 0.974 | 0.947 | 0.947 |
|    | Wu's Method [84] | - | **0.942** | 0.941 | 0.947 | - |
|    | Ours-full | **0.904** | 0.865 | 0.987 | 0.961 | 0.961 |
| WS | WSIS-BBTP [85] | 0.858 | 0.784 | 0.967 | 0.937 | 0.937 |
|    | Ours-weak | **0.874** | **0.861** | **0.986** | **0.950** | **0.950** |

**Table 3.** Performance comparison of the proposed segmentation network and other methods on ISIC 2017 dataset.

|    | Method | F1 | SEN | SPE | ACC | Jaccard Similarity |
|----|--------|-----|------|------|------|--------------------|
| FS | U-net [5] | 0.8682 | **0.9479** | 0.9263 | 0.9314 | 0.9314 |
|    | Melanoma det [86] | - | - | - | 0.9340 | - |
|    | Lesion Analysis [87] | - | 0.8250 | 0.9750 | 0.9340 | - |
|    | R2U-net [81] | 0.8920 | 0.9414 | 0.9425 | 0.9424 | 0.9421 |
|    | BCDU-Net [82] | 0.8810 | 0.8647 | 0.9751 | 0.9528 | 0.9528 |
|    | MCGU-Net [83] | 0.8950 | 0.8480 | 0.9860 | 0.9550 | 0.9550 |
|    | HRFB [88] | - | 0.870 | 0.964 | **0.938** | - |
|    | Deeplab v3+ [25] | 0.9162 | 0.8733 | **0.9921** | 0.9691 | 0.9691 |
|    | Att-Deeplab v3+ [79] | **0.9190** | 0.8851 | 0.9901 | **0.9698** | **0.9698** |
|    | Mask R-CNN [18] | 0.9092 | 0.8644 | 0.9794 | 0.9472 | 0.9472 |
|    | Wu's Method [84] | - | 0.9061 | 0.9628 | 0.9570 | - |
|    | Ours-full | 0.9145 | 0.8865 | 0.9879 | 0.9635 | 0.9636 |
| WS | Ours-weak | **0.8845** | **0.8473** | **0.9706** | **0.9384** | **0.9384** |

From these related works, performed for skin lesion segmentation, we can see that our method achieves a competitive performance compared with the classic lesion segmenting methods—Wu's method [84], HRFB [88] and Att-Deeplab V3+ [79]. In addition, our method is different to these related methods. Specifically: (1) The existing methods for skin lesion segmentation are the methods based on the fully supervised learning, while our proposed method can carry out lesion segmentation based on weekly supervised learning using box level annotations; (2) ADAM [84] attention module, which includes Global Average Pooling (GAP) and Pixel Level Correlation (PC), is designed in Wu's method to capture global contextual information. HRFB [88] provides high-resolution feature mapping to preserve spatial details. Att-Deeplab V3+ [79] introduces two levels of attention mechanism based on deeplab V3+ to capture the relationships between a group of features. We introduce the CBAM module into the FPN (Feature Pyramid Networks) network to form an attention FPN, so as to improve the network's perception of multi-scale images.

In summary, Figures 5 and 6 show the results of the comparison of the segmentation details between our method and other methods. It can be seen that after the attention mechanism is involved, the segmentation of lesion area will have fewer false segmentation and missing segmentation, and the segmentation of boundary details will be more accurate than the original network does. These qualitative results are exactly in line with our expectations of joining the dual attention mechanism, allowing the network to pay attention to "what" and "where". However, in the quantitative evaluation, our fully supervised method achieve the second best results regarding Att-Deeplab v3+ in the ISIC dataset, and surpassed other methods in all indicators on the oral leukoplakia dataset. However,

in the oral leukoplakia dataset, Att-Deeplab v3+ achieved the worst results. Even our weakly supervised segmentation results surpassed Att-Deeplab v3+. It can be seen that the segmentation framework we proposed can effectively extract the features of the lesion area; thus, it is robust and adaptive to different datasets. In the segmentation task of oral leukoplakia, due to the small amount of data in the oral leukoplakia dataset, the image size is extremely large, and the scale of the lesion area changes greatly, which will lead to the traditional feature extraction network to lose a lot of details after extracting higher-level information. If the lesion area is small, this area will be ignored, leading to missing segmentation. In contrast, our framework backbone network is based on the feature pyramid network of dual attention. The multi-scale network can fuse the high-level features with richer semantic information and the low-level features with higher resolution, and effectively reduce the phenomenon of missing segmentation. In the weakly supervised segmentation method, the detection model with the attention mechanism also greatly improved the ability of locating the lesion area, which provides an accurate bounding box for the segmentation of GrabCut [55].

In addition, we also analyzed the computational complexity of some related segmentation methods, and the results are shown in Table 4. From these results, we can see that our model have similar computational complexity with most segmentation methods.

**Table 4.** The computational complexity of related segmentation methods.

| Method | Params(M) | GFLOPs |
|---|---|---|
| U-Net [5] | 31 | 233 |
| R2U-Net [81] | 75 | 78 |
| Deeplab V3+ [25] | 59 | 67 |
| Attention U-Net [81] | 51 | 55 |
| Wu' method [84] | 38 | 33 |
| Our model | 44 | 47 |

## 5. Conclusions

In this paper, we propose an end-to-end medical lesion segmentation framework. In this framework, if pixel-level labels are available, we can use the fully supervised branch to obtain more precise segmentation results. If you only have box-level labels, you can still use the weakly supervised branch to obtain better segmentation results. In addition, we add a dual attention mechanism to improve the network segmentation performance. The dual attention mechanism in Mask R-CNN can help the network focus on the features of important regions, but suppress the unimportant features. This mechanism also provides a more accurate bounding box for weakly supervised branches. In addition, the proposed weakly supervised segmentation branch can greatly reduce the gap between labels and pseudo labels, and achieve comparable performance with fully supervised segmentation. Experimental results on the oral dataset and the ISIC 2018 dataset demonstrate the effectiveness of our proposed framework. In this paper, the fully and weakly supervision segmentation branches are used for lesion segmentation separately, rather than integrating into one model. Therefore, we can design an end-to-end weak supervision image segmentation model in the future.

**Author Contributions:** F.X. and P.Z. designed the research and wrote the manuscript; T.J., X.S., P.X., W.Z. and Z.G. contributed to the improvement of our ideas and to the revision of the manuscript; P.Z., J.S., X.S. and G.G. carried out the data collection and research experiments. All authors have read and agreed to the published version of the manuscript.

**Funding:** This research was funded by National Natural Science Foundation of China under grant agreements Nos. 61973250, 61973249, 61876145, 61802335, 61902313, 61802306. Shaanxi Province Science Fund for Distinguished Young Scholars: 2018JC-016. Key Research and Development Program of Shaanxi (Program No.2019GY-012, 2021GY-077, 2021ZDLGY02-06). Cross disciplinary Research Fund of Shanghai Ninth People's Hospital, Shanghai JiaoTong university School of Medicine (JYJC202113). Shaanxi Provincial Department of Education serves local scientific research: 19JC038.

**Informed Consent Statement:** Informed consent was obtained from all subjects involved in the study.

**Data Availability Statement:** The images in Oral Leukoplakia Dataset are collected from Department of Oral Mucosal Diseases, Shanghai Ninth People's Hospital, we have gotten the patients' consents before taking the images, and these images have no personal information of patients. ISIC 2018 is a public image dataset for medical image segmentation.

**Conflicts of Interest:** All the authors declare that we have no known competing financial interests or personal relationships that could have appeared to influence the work reported in this paper.

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
