# Peer review of "Lesion Segmentation Framework Based on Convolutional Neural Networks with Dual Attention Mechanism"

_electronics, doi:10.3390/electronics10243103_

Round 1
Reviewer 1 Report
In this paper, the authors proposed a lesion segmentation framework based on convolutional neural networks with dual attention mechanism. They used bounding-box labels instead of pixel-level labels for the segmentation of medical images at first. Then a dual attention mechanism, channel and spatial attention, was used to extract features from important regions. The experimental results showed that their proposed framework achieved better performances by comparing with several previous works on their oral lesion and ISIC2018 datasets.
Although the overall structure of the paper is complete, the method description is detailed, and the experimental results are good. However, the main innovative technology proposed in this paper has been proposed in other published papers, especially for the manuscript presented by Wu et al. (10.1109/TMI.2020.3027341). This paper does not cite this manuscript.
Therefore, in this paper, I did not see any new technology being proposed. I cannot compare the difference and effectiveness of this paper with the above paper. Therefore, this paper has the following problems.
1. Please cite the above-mentioned paper (Wu's paper), even some well-known papers have been ignored.
2. Please compare the differences of the technologies in details (Wu's paper and this paper), especially for the bounding-box labels and dual attention mechanism.
3. Please compare the experimental results predicted by Wu's paper and this paper in details.
4. The dataset proposed in this paper is too small (90 images, no test set). The statistical significance is not high. Moreover, it has not been rigorously reviewed by other experts. It is necessary to provide complete information+images for the dataset, and provide relevant experimental results predicted by this paper and others.
5. Authors should also use the famous skin image dataset (ISBI 2017) to do the experimental test.
6. The experimental results are insufficiently described, and relevant data should be further analyzed in order to explain the advantages and disadvantages of the proposed method.
Reviewer 2 Report
Comments are in the attachement

Reviewer 3 Report
This paper proposes an end-to-end CNN framework with dual attention mechanism for image lesion segmentation.
Interesting work, with promising results, but with important deficiencies in the introduction of the methods and doubts in the validity of the comparative studies.
Methodological issues
- the manuscript lacks a clear structure
- work contributions are not easy to identify or understand
- the description of the attention modules is insufficient and confusing
- the wording is often very dense and very difficult to follow
Authors are asked to make an effort aimed at simplifying linguistic constructions and providing a better structure for the technical presentation of the method, establishing clear boundaries between the architecture inherited from the state of the art and the contributions made. In this regards, the "Materials and Methods" section could not encompass any further review of the state of the art, and could focus on presenting the methods and data involved.
Technical issues:
- Line 198: "In CNN, the depth (receptive field) of the network and the step size are opposite"; what did you mean?
- What is the difference between the two schemes in Fig. 3?
- No images from the oral leukoplakia dataset were used for fair/objective test purposes. How were the hyperparameter values selected / optimized?
- In the comparative study involving the oral leukoplakia dataset, how were the results of the other methods obtained? Were they implemented from scratch? Were their hyperparameters independently optimized for each database?
- In the case of the distribution of the ISIC 2018 images for training, validation and test purposes, is it exactly the same as in the rest of the methods included in the comparison?
A thorough linguistic revision of the manuscript is necessary.
Please check the wording of these sentences:
- While they also have challenges to object segmentation.
- As shown in Figure 1.
- And provides a network structure that combines high-level semantic features with low-level features to generate high rate segmentation graphs.
- ...which is supplement to channel attention.
Typos
- line 101: In this paper, The researchers
- Fig. 2: 'Segmentatio'
- line 186: labelsThisimproves
- line 228: the acronym MLP is used before its definition
- line 234: ...\alpha represents... (it should be \sigma)
- eq. (2): F_arg should be F_avg
- line 255: N represents the number of datasets (samples?, examples?, instances?)
- line 288: finetune
- line 306: the Ground-Truth labels (capital letters are not necessary)
Round 2
Reviewer 1 Report
The authors have answered related questions and suggestions. No additional question or suggestion in the revised version of this manuscript.
Reviewer 2 Report
My all comments and suggestions have been addressed. Thanks
Reviewer 3 Report
The authors have satisfactorily addressed the issues indicated in the first round of the review. A final linguistic revision of the manuscript is recommended.